# Ciliary Neurotrophic Factor (CNTF) Inhibits In Vitro Cementoblast Mineralization and Induces Autophagy, in Part by STAT3/ERK Commitment

**DOI:** 10.3390/ijms23169311

**Published:** 2022-08-18

**Authors:** Jiawen Yong, Sabine Gröger, Julia von Bremen, Sabine Ruf

**Affiliations:** 1Department of Orthodontics, Faculty of Medicine, Justus Liebig University Giessen, 35392 Giessen, Germany; 2Department of Periodontology, Faculty of Medicine, Justus Liebig University Giessen, 35392 Giessen, Germany; 3Stomatology Hospital, School of Stomatology, Zhejiang University School of Medicine, Zhejiang Provincial Clinical Research Center for Oral Diseases, Key Laboratory of Oral Biomedical Research of Zhejiang Province, Cancer Center of Zhejiang University, Hangzhou 310003, China

**Keywords:** cementoblasts, CNTF, cementogenesis, autophagy, ERK1/2, STAT3

## Abstract

In animal models, the administration of ciliary neurotrophic factor (CNTF) was demonstrated to reduce bone mass and to participate in bone remodeling. Cementoblasts, a cell type embedded in the cementum, are the main cells to produce and mineralize the extracellular matrix. The effect of CNTF on cementoblasts has not yet been addressed. Thus, the goal of this in vitro study was to investigate possible influences of exogenous CNTF on cementogenesis, as well as autophagy regulation and subsequent mechanisms in cementoblasts. Cementoblasts (OCCM-30) were stimulated with exogenous CNTF. Alizarin Red staining was performed to analyze the functional differentiation (mineralization) of OCCM-30 cells. The release of OPG was quantified by ELISA. The expression of cementogenesis markers (*RUNX-2*, *OCN*, *BMP-7*, *BSP*, and *SPON-2*) was evaluated by RT-qPCR. Western blotting (WB) was performed for the protein expression of STAT3, COX-2, SHP-2, cPLAα, cPLAβ; ERK1/2, P38, and JNK. The autophagic flux was assessed using WB and RT-qPCR analysis of LC3A/B, Beclin-1, and Atg-5, and the autophagosome was investigated by immunofluorescence staining (IF). The ERK1/2 (FR180204) or STAT3 (sc-202818) antagonist was added, and the cellular response was analyzed using flow cytometry. Exogenous CNTF significantly attenuated mineralized nodule formation, impaired OPG release, and downregulated the mRNA levels of *RUNX-2*, *OCN*, *BMP-7*, and *BSP*. Moreover, CNTF induced the phosphorylation of STAT3 and activated a transient activation of SHP-2, cPLAβ, ERK1/2, P38, and JNK protein. CNTF also induced autophagosome formation and promoted autophagy-associated gene and protein expressions. Additionally, the inhibition of ERK1/2 or STAT3 reversed a CNTF-induced mineralization impairment and had regulatory effects on CNTF-induced autophagosome formation. Our data revealed that CNTF acts as a potent inhibitor of cementogenesis, and it can trigger autophagy, in part by ERK1/2 and STAT3 commitment in the cementoblasts. Thus, it may play an important role in inducing or facilitating inflammatory root resorption during orthodontic tooth movement.

## 1. Introduction

The ciliary neurotrophic factor (CNTF) is one of the neurocytokines that signals through a ciliary neurotrophic factor receptors complex and has been shown to play a pivotal role in bone remodeling, as well as in the promotion of nerve cell survival [1]. It belongs to the interleukin (IL)-6 related cytokine family [2], which includes oncostatin M and IL-6 [3]. CNTF is widely expressed in the brain, spinal cord, and ciliary ganglia [4]. However, the expression and concentration of CNTF in gingival crevicular fluid has not been detected. CNTFR expression has been described in osteoblast-like cells [5]. It plays a role in the inhibition of CNTF in osteoblasts in vitro and also suppresses the trabecular bone metabolism [1]. When bound to CNTF, CNTFR forms a signaling complex with gp130, which plays a critical role in bone cell biology [5]. 

Cementoblasts, highly differentiated mesenchymal cells of the periodontal ligament located in the cementum, mediate the periodontal tissue remodeling, especially cementum regeneration, during orthodontic tooth movement (OTM) [6]. In the cementum, the extracellular matrix (ECM) is produced. This is composed of collagenous proteins and non-collagenous proteins, such as runt-related transcription factor-2 (RUNX-2), osteocalcin (OCN), bone sialoprotein (BSP) [7], bone morphogenetic protein-7 (BMP-7) [8], and osteoprotegerin (OPG). It maintains the cementoblastic homeostasis and promotes cementum regeneration. We recently reported about the expression and location of the ciliary neurotrophic factor receptors complex in cementoblasts [9]. This pivotal finding led us to investigate the possible regulatory effects of CNTF in cementoblast differentiation, especially for mineralization.

Autophagy, an important cellular adaptation mechanism, has already been demonstrated to be an important process during OTM [10]. It is a self-digestive intracellular process that degrades cytoplasmic components by lipid double-membrane structures (i.e., autophagosomes) for delivery to lysosomes or vacuoles for degradation [11]. It serves as an environmental adaptation reaction and is upregulated under diverse pathological conditions, such as external stress, starvation, hypoxia, immune response, and bacterial infection [12]. Autophagy occurs along with microtubule-associated protein light chain 3 (LC3)-A/autophagosome-associated LC3-B conversion [13], Beclin-1, and evolutionarily conserved autophagy-related (ATG) gene mediation. Recent studies have shown that, under an inflammatory microenvironment, autophagy is activated and exhibits protective effects in periodontal ligament stem cells [14], as well as in pre-odontoblastic cells [15]. Nollet et al. (2014) showed that an autophagy deficiency reduced the mineralization capacity, and autophagy-deficient osteoblasts exhibited an increased oxidative stress and secretion of NF-kB/RANKL, favoring the generation of osteoclasts. In vivo, they observed a 50% reduction of autophagy in bone mass from osteoblast-specific autophagy-deficient mice. Therefore, the roles of autophagy in periodontal tissue and the pathogenesis of orthodontically induced inflammatory root resorption (OIIRR) [16] during OTM have been a research focus [16]. More molecular and cellular investigations on cementoblasts are needed to elucidate the underlying mechanism that governs the regulation of the mineralization and autophagy pathway.

Since the mechanism of cementogenesis and autophagic regulation by exogenous CNTF remains unclear in cementoblasts, the aim of this in vitro study was to evaluate the potential regulatory effects of CNTF on the biological responses of cultured mouse cementoblasts, including the regulation of mineralization and autophagy.

## 2. Results

### 2.1. CNTF Impairs Cementoblast Mineralization and Impairs the Expression of Cementogenesis Hallmarkers

First, the sustained stimulation of exogenous CNTF on the extracellular matrix production, as well as the OPG activity of cementoblasts, was examined by Alizarin Red Staining and ELISA. We observed that OCCM-30 cells cultivated in the presence of CNTF (400 ng/mL) showed a markedly reduced intensity of mineralization (42.33 ± 8.54%) in comparison with the control group (Figure 1A,B). As colorimetric analysis showed, exogenous CNTF impaired OPG enzymatic activity (Figure 1C). Interestingly, this inhibition was time dependent, and it appeared that the inhibitory effect of CNTF on OPG is significant within 90 min of stimulation (Figure 1C).

To further investigate the findings that exogenous CNTF inhibits mineralization, we performed RT-qPCR to analyze the expressions of key cementoblast signaling transcription molecules involved in cementoblast mineralization in the presence or absence of CNTF. The quantitative mRNA expression analysis showed drastically downregulated levels of *RUNX-2* (0.13 ± 0.05 folds), *OCN* (0.70 ± 0.09 folds), *BMP-7* (0.63 ± 0.20 folds), and *BSP* (0.73 ± 0.14 folds) by CNTF stimulation in the cementogenesis medium, as compared to control cells grown in the same medium (Figure 1C). No significant changes in the mRNA expression of *SPON-2* were demonstrated after exogenous CNTF stimulation.

### 2.2. CNTF Regulates STAT3 as Well as COX-2, SHP-2, and cPLAβ Expression in OCCM-30 Cells

Next, we stimulated OCCM-30 cells with CNTF and performed Western blot analysis assessing the kinetics (Figure 2). Cells stimulated with CNTF (400 ng/mL) show upregulation of phosphorylated STAT3 after CNTF addition at 5 min, followed by a decrease in phosphorylation levels 45 min after stimulation (Figure 2A,B). Interestingly, cells showed inactivation of COX-2 that occurred from 15 to 120 min of CNTF stimulation (Figure 2C,D). A strong expression level of SHP-2 was detected, beginning after 15 min until 3 h in response to CNTF stimulation. A constant expression of cPLAβ was found after 5 min of CNTF stimulation, whereas no effect on the expression level of cPLAα was observed (Figure 2E,F).

### 2.3. CNTF Triggers ERK1/2, P38, and JNK Phosphorylation in Cementoblasts

To elucidate whether or not CNTF can activate the MAPK pathway, we performed a kinetic analysis of ERK1/2, P38, and JNK protein phosphorylation by Western blotting (Figure 3).

Western blots demonstrated that the phosphorylated state of ERK1/2 occurred 5 min after CNTF (400 ng/mL) stimulation. The ERK1/2 phosphorylation was sustained over a period of 1 h, reaching a peak after 15 min (Figure 3A,B). The phosphorylation of P38 and JNK reached a peak after 15 min of CNTF addition and was detectable over a period of 2 h (Figure 3C–F).

### 2.4. CNTF Induces the Autophagosome and Activates Autophagic Signaling

In order to determine the influence of CNTF as an autophagosome inductor on cementoblasts, the autophagy regulation was determined by IF staining (Figure 4A). We observed that in the group of cells stimulated with CNTF, the fluorescence intensity was significantly increased (Figure 4A), which suggested a higher incidence of autophagosomes induction and an enhancement of the autophagic flux. On cells cultivated under regular growth medium in the absence of CNTF, the fluorescence intensity was remarkably low (Figure 4A). At the gene level, CNTF stimulation for 32 h significantly increased the autophagy-associated gene expression of *LC3A*, *Beclin-1,* and *Atg-5* (Figure 4B).

Furthermore, we performed Western blotting for autophagy-associated proteins stimulated with CNTF (400 ng/mL) (Figure 4C,D). The immunoblotting showed that CNTF stimulation increased not only processed LC3B, but also LC3A expression from 2 to 24 h, which indicates an elevation of the autophagic flux (Figure 4C,D). The same upregulating effects were observed in the Beclin-1 as well as Atg-5 protein expression after 1 to 32 h of CNTF stimulation (Figure 4C–E).

### 2.5. STAT3/ERK Are Involved in CNTF-Induced Inhibitory Cementoblast Mineralization Effects and Autophagy Induction

Finally, to determine if STAT3 and ERK1/2 play a regulatory role in the process of mineralization and autophagy in response to CNTF stimulation, we conducted pharmacological inhibition experiments using two molecules, a specific inhibitor for STAT3 (sc-202818) and one for ERK1/2 (FR180204), to determine their involvement.

The Alizarin Red staining revealed that STAT3 inhibition significantly increased the mineralization of cementoblasts at 14 days, whereas CNTF reversed this effect (Figure 5A,B). Meanwhile, in the presence of CNTF, the blockade of STAT3 or ERK1/2 significantly increased the OPG release level to 2216.60 ± 87.98 pg/mL and 2104.94 ± 46.41 pg/mL, respectively (Figure 5C). The co-stimulation of CNTF and sc-202810 or FR180204 showed a significant downregulation in the OPG release (Figure 5C).

Furthermore, autophagy regulation was determined by flow cytometric analysis when co-cultivated with DMSO-only, sc-202818, or FR180204 in the presence or absence of CNTF treatment (Figure 5D,E). The fluorescence intensity of cells treated with sc-202818 was shown to be significantly decreased (Figure 5E). The ERK1/2 antagonist FR180204 also inhibited the fluorescence intensity, whereas the CNTF addition reversed this effect, suggesting a lower incidence of autophagosomes (Figure 5E).

## 3. Discussion

In the current study, we present data that revealed for the first time that CNTF reduces in vitro the mineralization of cementoblasts and decreases the OPG release by activating ERK1/2 and STAT3 signaling. We could also show that CNTF induces autophagy via LC3B, Beclin-1, and Ayg-5 signaling. In this study, a novel mechanism demonstrated that STAT3 and ERK1/2 participate in the CNTF-regulated cementogenesis and autophagy. This may influence the process of OIIRR during OTM.

To the best of our knowledge, this is the first study providing evidence that CNTF could interact with cementoblasts and regulate their biological response during cementogenesis. One of the main roles of cementoblasts during OIIRR is to produce and mineralize ECM [17] in order to regulate cementum regeneration [7]. The study of Johnson et al. (2014) reported that CNTF suppresses the gene expression of osteoblastogenic markers, such as *RUNX-2*, *Osterix* (*OSX*), *Alkaline Phosphatase* (*ALP*), and *OCN* in murine calvarial osteoblasts [18]. Another work by McGregor et al. (2010) demonstrated that female CNTF-deficient mice exhibited higher osteoblast numbers and increased mineralization, indicating that CNTF inhibits ECM in osteoblasts and plays an inhibitory role regaring trabecular bone formation in female mice [5]. In this study, we showed that CNTF reduces OPG secretion, as well as the cementogenesis marker genes expression in cementoblasts, suggesting that CNTF impairs the mineralization of cementoblasts. Taken together, CNTF, a cytokine of the IL-6 family, exerts in vitro anti-cementogenesis effects on cementoblasts.

Previous studies indicate that CNTF can induce ERK1/2 and STAT3 activation in chicken nodose neurons [19]. We found that CNTF could interact with cementoblasts in a similar way (see Figure 1). In the present report, we observed that CNTF strongly induces ERK1/2 phosphorylation, an effect that suggests that ERK1/2 MAPK signaling activation by CNTF occurs time-dependently in cementoblasts. However, phosphorylated P38 and JNK induced by CNTF were observed for a short time period. Furthermore, we observed that exogenous CNTF strongly induced the tyrosine (Tyr705) phosphorylation of STAT3 in cementoblasts in a time-dependent manner. These findings indicate that ERK1/2 and STAT3 commitments may participate in the homeostasis regulation of cementoblasts.

The roles of STAT3 and MAPK signaling in autophagy and mineralization in cementoblasts was recently highlighted by several studies [7,20,21,22]. Depending on the cell type or stimulus, activated ERK1/2 or STAT3 catalyzes several cell processes, including differentiation and autophagy. Wang et al. (2018) showed that TNF-α-induced autophagy in cementoblasts was dependent on the STAT3 signaling pathway [20]. Very recent research by Ma et al. (2021) revealed that co-culture with *P. gingivalis* enhanced autophagy activity in cementoblasts, which was partially mediated by Jak/STAT3, PI3K-Akt, and ERK1/2 signaling [21]. We thus conducted pharmacological inhibition experiments to clarify the mechanisms. Finally, we observed that exogenous CNTF added to the cell cultures impairs OPG release. At the same time, the secretion of OPG was efficiently counteracted by ERK1/2 or STAT3 blockade. Thus, this effect is regulated by ERK1/2 and STAT3 commitment. This indicates that the pharmacological blockade of both molecules seems to have a protective effect on cementoblasts against CNTF-induced cementogenesis inhibition. 

Autophagy is an important protective cellular process to maintain cell homeostasis under inflammatory conditions [22]. However, it is unclear whether and how CNTF influences autophagy. Our results showed that CNTF could induce autophagosome formation and the time-dependent effects of CNTF stimulation upon raising the levels of LC3B, Beclin-1, and Atg-5 proteins and genes. Therefore, we provide conclusive evidence that exogenous CNTF can induce autophagy in cementoblasts. Furthermore, CNTF activity is blocked by sc-202818, indicating that STAT3 molecules are implicated in autophagy induction in cementoblast.

Moreover, recent publications provide evidence that autophagy is involved in mineralization [23]. Nollet et al. (2014) demonstrated that the autophagy of osteoblasts is involved in the mineralization process and in bone homeostasis [24]. Vrahnas et al. (2019) reported that autophagic processes in osteocytes may directly control mineralization and modify bone quality by the regulation of mineral secretion [23]. Using an in vitro mouse model, it was demonstrated that the inhibition of autophagy during OTM increased the inflammation-related gene expression and decreased alveolar bone density, a process that accelerates OTM [25]. In the present study, we observed that inducible autophagy and decreased mineralization capacity occurred in response to CNTF stimulation in cementoblasts, indicating that the mineralization process is possibly associated with the induction of autophagy. Bafilomycin A1 (BA1) and chloroquine (CQ) are two commonly used compounds that inhibit autophagy by targeting the lysosomes, but through distinct mechanisms [26]. BA1 blocks the fusion of the autophagosome with the lysosome, as well as the subsequent acidification, and CQ blocks the final step of degradation in the autolysosome [27] in MC3T3-E1 cells. These two chemical reagents served as autophagy inhibitor, but were not fully investigated in the cementogenesis process of cementoblasts until now. Thus, this interesting hypothesis requires future studies, including research regarding the inhibition of autophagy by these two pharmacological inhibitors, in order to confirm the physiological effects of autophagy in the cementogenesis of cementoblasts in vitro. Additionally, this in vitro study was carried out by using a murine cementoblast cell line. In future studies, animal experiments will be useful to evaluate the revealed mechanisms in vivo, prior to investigation in humans.

## 4. Materials and Methods

### 4.1. Cell Culture and Conditions

The immortalized murine cementoblast cell line (OCCM-30) [28,29] (generously provided by Prof. J. Deschner and Dr. M. Nokhbehsaim, Department of Periodontology, University of Bonn, Germany) was plated with a seeding density of 1 × 10^6^ cell/well and maintained at 37 °C in a humidified atmosphere with 5% CO_2_ in air. OCCM-30 cells were cultured in the regular growth medium α-MEM (Gibco, Gaithersburg, MA, USA), supplemented with 10% (*v*/*v*) heat-inactivated fetal bovine serum (FBS) (Gibco) and 1% penicillin/streptomycin (Gibco) in 6-well plates (Greiner bio-one, Frickenhausen, Germany). The growth medium was replaced every day with fresh medium. Until grown of adherently cementoblasts to 80% confluency, the concentration of FBS was set to 0.5% as a starvation medium 14 h prior to CNTF stimulation [30]. All experiments were performed using cells at the 2nd to 5th passages.

### 4.2. Differentiation (Mineralization) Induction

In order to induce cementoblastic differentiation (mineralization), the cementogenesis induction medium (α-MEM with 10% FBS, supplemented with 10 mM β-glycerophosphate (Sigma, Taufkirchen, Germany)) and 50 μg/mL ascorbic acid (Sigma) were added to the culture. The cementogenesis medium was changed every 48 h during the mineralization induction period [7].

### 4.3. Reagents and Pharmacological Inhibitor

OCCM-30 cells were kinetically stimulated using recombinant mouse CNTF protein (Novus Biologicals, Wiesbaden, Germany). Additionally, mouse recombinant interleukin-6 (IL-6) (Sigma) was applied as a positive control for cementogenesis assay.

For the signaling inhibition experiments, the STAT3 inhibitor V (sc-202818, Santa Cruz Biotechnology, Heidelberg, Germany) as well as an ERK1/2 inhibitor (FR180204, Calbiochem, CA, USA) were used at 1.0 μg/mL, both dissolved in dimethyl sulfoxide (DMSO) (Sigma), and the same amount of DMSO was used as negative control group.

### 4.4. Alizarin Red Assay

The induction of mineralization by means of mineral deposition amount detection was measured using Alizarin Red staining, as previously described [31]. Briefly, the extracellular matrix was washed with ice-cold 1 × phosphate-buffer saline (PBS), fixed with fixation buffer (BD Cytofix^TM^, Thermo Fisher Scientific, Leipzig, Germany) and stained with 2% Alizarin Red solution (Sigma) for 5 min. Any nonspecific excess staining was thoroughly removed by washing with deionized water (Sigma). Photographs were obtained with a Leica DMI6000 B microscope (Leica Inc., Wetzlar, Germany) to assess the mineralized nodule formation.

Alizarin Red was eluted using 100 mM cetylpyridinium chloride (Sigma) to allow for releasing calcium-bound dye into the solution. The Alizarin Red optical density (O.D.) was measured on a spectrophotometer (BioTek, Winooski, VT, USA) at a wavelength of 490 nm and normalized to the control group.

### 4.5. Enzyme-Linked Immunosorbent Assay (ELISA)

OCCM-30 cells were seeded in 24-well culture plates at a density of 5 × 10^3^ cells/well until 80% confluency and then cultured in cementogenesis induction medium, as described above. After 2 h of pre-treatment with pharmacological inhibitors, the cells were then stimulated with CNTF. After the indicated time period incubation, the culture medium was collected for ELISA measurement.

OPG levels on cell supernatants were measured using a highly sensitive mouse Osteoprotegerin/TNFRSF11B ELISA kit (mouse) (P249109, R&D systems, Minneapolis, MN, USA), as previously described [32]. The optical density of each well was read at 450 nm with a 540 nm correction wavelength set, and the results were analyzed with Excel software (Microsoft, Washington, DC, USA) based on the standard curve of the protein of interest.

### 4.6. Gene Expression Analysis

Total RNA was isolated from CNTF treated-cells using a NucleoSpin^@^ RNA Kit (Macherey-Nagel, Dueren, Germany) according with the manufacturer’s instructions. The RNA concentration in each sample was quantified using a NanoDrop 2000 Spectrophotometer (Thermo Fisher Scientific). For cDNA synthesis, 1 μg of total RNA was reversed-transcribed to yield cDNA by using iScript^TM^ cDNA Synthesis Kit (Bio-Rad Laboratories, Feldkirchen, Germany).

Real-time qPCR amplification was performed using SsoAdvanced^TM^ Universal SYBR Green Supermix (Bio-Rad Laboratories) in a CFX96^TM^ Real-Time System (C1000^TM^ Thermal Cycler, Bio-Rad Laboratories). Primers were designed by Bio-Rad Laboratories, including *RUNX-2* (qMmuCID0005205, genomic sequence: NC_000083.6), *OCN* (qMmuCED0041364, genomic sequence: NC_000069.6), *BMP-7* (qMmuCID0007070, genomic sequence: NC_000068.7), *BSP* (qMmuCID0006396, genomic sequence: NC_000071.6), *SPON-2* (qMmuCED0048198, genomic sequence: NC_000071.6), *LC3A* (qMmuCED0045817, genomic sequence: NC_000068.7), *Beclin-1* (qMmuCED0049508, genomic sequence: NC_000077.6), and *Atg-5* (qMmuCED0048259, genomic sequence: NC_000076.6).

The temperature protocol of RT-qPCR was applied and comprised the following procedures: initial denaturation of cDNA at 95 °C for 30 s, followed by 40 cycles of combined annealing/extension procedure at 95 °C for 15 s and 60 °C for 30 s per cycle. The relative gene expression was normalized to the *PPIB* (qMmuCED0047854) [33] expression level and calculated using the 2^−ΔΔCq^ (quantification cycle) method to determine the fold difference. ΔC_q_ values = C_q_ of target gene—C_q_ of *PPIB*. ΔΔC_q_ = ΔC_q_ of target in stimulated-cells—ΔC_q_ of the same gene in unstimulated-cells.

### 4.7. Protein Expression Analysis

Whole cell extracts were collected using ice-cold lysis RIPA-buffer (Thermo Fisher Scientific) containing a 3% mixture of phosphatase and protease inhibitor cocktail (Thermo Fisher Scientific). A total amount of 20 μg per lane total protein lysates was separated by sodium dodecyl sulphate-polyacrylamide gel (SDS-PAGE) electrophoresis on 4–20% Mini-PROTEAN Precast Gel (Bio-Rad Laboratories) and transferred onto nitrocellulose membranes (Bio-Rad Laboratories).

Membranes were then blocked with 1 × Tris-buffered saline containing 5% (*w*/*v*) non-fat milk powder (Carl Roth, Karlsruhe, Germany) with 0.05% (*v*/*v*) Tween-20 (Merck, Darmstadt, Germany) at RT for 1 h. To detect the specific antigens, the appropriate primary antibodies were used, as follows: anti-p-STAT3 (Tyr705) (#9131, CST, Frankfurt a. Main, Germany), STAT3 (#9139, CST), COX-2/Cyclooxygenase 2 (ab62331, Cambridge, UK), SHP-2 (PA5-27312, Thermo Fisher Scientific), cPLAα (orb100010, BIOZOL, Eching, Germany), cPLAβ (ab198898, Abcam, Cambridge, UK ), Phospho-p44/42 MAPK (ERK1/2) (Thr202/Tyr204) (#4370, CST), p44/42 MAPK (ERK1/2) (#4695, CST), p54/p56 JNK (#9252, CST), phosphor-JNK (Thr183/Tyr185) (#4671, CST), P38 MAPK (#9212, CST), phospho-P38 (Thr180/Tyr182) MAP (#4511, CST), LC3A/B (#12741, CST), Beclin-1 (#3495, CST), and Atg5 (#12994, CST) at dilutions of 1:1000. The standardize protein loading was controlled by β-actin (ab8227, Abcam) at a dilution of 1:2500. The respective HRP-conjugate polyclonal Goat Anti-Rabbit (P0448, Dako, Santa Clara, CA, USA) and Goat Anti-Mouse (P0447, Dako) were subsequently incubated as the secondary antibody at a dilution of 1:2000.

Visualization of the immunoreactive proteins were detected by enhanced chemiluminescence (Amersham ECL Western Blotting Detection Reagents, GE Healthcare Life Sciences, Tokyo, Japan) by means of image lab software (Bio-Rad Laboratories). A free image-processing ImageJ software 2.0 (National Institutes of Health, Washington, DC, USA) was used to perform the densitometrical analysis of the bands.

### 4.8. Autophagy Detection (Autophagosome Quantification)

Following CNTF stimulation, autophagosomes were stained using the Cyto-ID^@^ Autophagy Detection Kit (ENZ-51031, Enzo Life Sciences, Exeter, UK) for immunofluorescence (IF) staining and flow cytometric analysis [6,7]. The Cyto-ID^@^ 488 nm-excitable green fluorescent detection reagent accumulates specifically in the autophagosomes in a pH-dependent manner [34]. The IF staining procedure was conducted according to the manufacturer’s instructions and comprised a positive control with the autophagy inducer starvation incubation. Unstained cells served as a negative control. In short, the monolayer cells were carefully washed with 1 × washing buffer containing 5% FBS and incubated with the Cyt-ID^@^ Dual Detection Reagent for 30 min in the dark at 37 °C. After removing excess buffer, samples were fixed with the fixation buffer (BD Cytofix^TM^). The autophagic signal (FITC filter set) and the nuclear signal (DAPI filter set) were photographed using a high-resolution fluorescence microscope (Leica) with 60× magnification. Additionally, flow cytometry was performed with the FACS SP6800 Vantage Flow Cytometer (Sony Biotechnology, Berlin, Germany) at 530/30 nm excitation for FITC, following analysis using the FlowJo software 1.0 (BD Biosciences, CA, USA).

### 4.9. Statistical Analysis

GraphPad Prism 8.0 software (GraphPad Software Inc., San Diego, CA, USA) was used for statistical analysis. Quantitative data analysis is determined as means ± standard deviation (SD) and applied to the analysis using independent Student’s *t*-tests to compare between two groups. Statistical significance between groups was considered when *p* value was <0.05. Each in vitro experiment was performed in triplicate and reproduced at least twice.

## 5. Conclusions

In conclusion, our data show that, in addition to its deleterious effect on cementoblasts, impairing the mineralization as well as the release of OPG during cementogenesis, CNTF also has regulatory effects on autophagy induction. Furthermore, it was revealed that CNTF regulates cementoblast homeostasis, in part by the ERK1/2 and STAT3 signaling network.

## Data Availability

The datasets used and/or analyzed during the current study are available from the corresponding author on reasonable request.

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
