# Peer review of "Ciliary Neurotrophic Factor (CNTF) Inhibits In Vitro Cementoblast Mineralization and Induces Autophagy, in Part by STAT3/ERK Commitment"

_ijms, 2022, doi:10.3390/ijms23169311_

Round 1

Reviewer 1 Report

The current manuscript has been reviewed, and the manuscript needs changes before it can be further processed.

There is a high need for improvisation of images such as IF (figure 4) and 1 ex: micrograph.

The general workflow of the experiment is apt. 

Reviewer 2 Report

I have read the presented manuscript. The paper is globally well-written and well-presented. Results are clearly discussed. However, there are some minor and major problems that need to be solved.

Introduction: I have nothing to report in this part. 

Materials and methods: Please specify in line 100-101 the incubation period or, at least, a reference that reports this information. The extensive name of DMSO in line 109 presents a mistake. In the 2.6 Gene expression analysis paragraph the gene accession number of the analyzed genes is missing. Moreover, I suggest to change Cq in Ct

Results: I suggest to fill the histogram's bars in figure 1C since it is not clear. Figure 1D shows a massive decrease in the mRNA levels of several genes. Why the Authors did not investigate also the protein levels? 

The quality of figure 4A is very low. It is not possible to distinguish anything. Authors have to provide a new image of better quality. The densitometric analysis of figures 4C and 4D is missing. Why the Authors did not study the autophagy flux using Bafilomycin A1/cloroquin? What happens to the mineralization process (e.g., Alizarin Red assay) after the treatment of cementoblasts in presence/absence of autophagy inhibitors?

It is not clear if figure 5B refers to the 7 days treatment with STAT3 and Erk1/2 inhibitors or to the 14 days of treatment. The Authors reported that STAT3 or Erk1/2 inhibition led to an increased mineralization in cementoblasts (they do not specify the investigated time interval. Is 7 days or 14 days p.t.?) and that this effect is reversed by CNTF. Figure 5A seems not to support this explanation. After 7 days it is clear that CNTF decrease the mineralization process in cementoblasts, as already reported in figure 1A. Among the two investigated inhibitors (i.e., sc-202818 and FR180204) it is evident that sc-202818 led to a decrease in the mineralization process as in the case of CNTF. The co-treatment led to a further decrease in mineralization. Same consideration can be made for FR180204, even if the relative contribution of this compound is less compared to sc-202818. Furthermore, it seems that the effect exerted by all the compounds wore off after 14 days p.t. Why the Authors have not included the analysis of OD of Alizarin Red? It is also reported in figure 5B that both the inhibitors led to an increase in OPG levels (at 7 days? 14 days?) but CNTF led to a decrease of this factor. What are the levels of OPG in the combined treatment (i.e., CNTF + sc-202818 or CNTF + FR180204)? Figures 5C and 5D are not clear. Firstly, Authors should include a the panel showing authophagy levels in presence of CNTF, as control. Another conclusion is that CNTF enhanced the autophagy inhibition exerted by sc-202818 but there no significant difference between the two histograms. How can the Authors conclude this? I suppose there is also a mistake in the + and - of the figure 5D regarding the sc-202818 treatment. Conversely, CNTF led to an increase in authophagy levels in presence of FR180204. How can the Authors explain this difference? How can the Authors be sure of the flow cytometry results, considering that the fluorescent signals presented in figure 4A was very faint? 

Discussion: The Authors proposed the hypothesis that CNTF inhibits the process of mineralization in cementoblasts and that this effect is mediated by Erk1/2 and STAT3 signalling pathway. Considering the results in figure 5A and 5B it is clear that CNTF leads to a decrease in mineralization process also through other mechanisms, as highlighted by the levels of Alizarin Red in the combined treatments. Furthemore, it is clear that the changes observed in the autophagy pathway can be address to STAT3 instead of Erk1/2, since CNTF reversed the autophagy inhibition promoted by FR180204. As asked above, in the presented paper the Authors have not investigated the effects of CNTF on mineralization in presence/absence of autophagy inducers/inhibitors. Why? This kind of experiments could be useful to link the autophagy pathway with the mineralization process in cementoblasts.

Round 2

Reviewer 1 Report

The current version has been improvised and the changes have been amended. 

Author Response

Thank you very much for the improvement in the manuscript quality.

Reviewer 2 Report

The Authors have improved their manuscript according to my observations. 

I still have only two doubts:

1. Figure 4E shows the densitometric analysis of LC3-II. I see a huge Standard Deviation and a low number of replicates (n=3). Generally, it is very hard to obtain a statistical significance with these premises. 

2. In the Response to Reviewer, at point 8, the Authors stated that Erk1/2 is implicated the autophagy's regulation in cementoblasts since CNTF increased the levels of autophagosomes in presence of FR180204. This interpretation is wrong. CNTF regulates autophgay mainly through STAT3 instead of Erk1/2. In fact, CNTF activity is blocked by sc-202818 (suggesting that STAT3 is implicated in autophagy induction) but not by FR180204 (suggesting that CNTF can induce autophagy regardless of Erk1/2 activation).   
